# Transformers and large language models are efficient feature extractors for electronic health record studies

Kevin Yuan[1,8] ✉, Chang Ho Yoon[1,8], Qingze Gu[2,8], Henry Munby[3], A. Sarah Walker[2,4,5], Tingting Zhu [6] & David W. Eyre [1,4,5,7]

## Abstract

**Background** Free-text data is abundant in electronic health records, but challenges in accurate and scalable information extraction mean less specific clinical codes are often used instead.

**Methods** We evaluated the efficacy of feature extraction using modern natural language processing methods (NLP) and large language models (LLMs) on 938,150 hospital antibiotic prescriptions from Oxfordshire, UK. Specifically, we investigated inferring the type(s) of infection from a free-text "indication" field, where clinicians state the reason for prescribing antibiotics. Clinical researchers labelled a subset of the 4000 most frequent unique indications (representing 692,310 prescriptions) into 11 categories describing the infection source or clinical syndrome. Various models were then trained to determine the binary presence/absence of these infection types and also any uncertainty expressed by clinicians.

**Results** We show on separate internal ($n = 2000$ prescriptions) and external test datasets ($n = 2000$ prescriptions), a fine-tuned domain-specific Bio+Clinical BERT model performs best across the 11 categories (average F1 score 0.97 and 0.98 respectively) and outperforms traditional regular expression (F1 = 0.71 and 0.74) and n-grams/XGBoost (F1 = 0.86 and 0.84) models. A zero-shot OpenAI GPT4 model matches the performance of traditional NLP models without the need for labelled training data (F1 = 0.71 and 0.86) and a fine-tuned GPT3.5 model achieves similar performance to the fine-tuned BERT-based model (F1 = 0.95 and 0.97). Infection sources obtained from free-text indications reveal specific infection sources 31% more often than ICD-10 codes.

**Conclusions** Modern transformer-based models have the potential to be used widely throughout medicine to extract information from structured free-text records, to facilitate better research and patient care.

## Plain language summary

Electronic health records often contain detailed information on clinical decisions and patient histories that are written as free text and otherwise not recorded in a structured format in a specific section of the record. Extracting specific information from this unstructured text is challenging, leading to researchers often using less detailed clinical information. This study evaluated whether computational methods, including large language models, could be used to extract detailed information from unstructured sections of medical records. As an example task, we attempted to identify which type of infection was being treated from free-text justifying antibiotic prescriptions. We could categorise infection types more often and more accurately than previous methods. This method of extracting detailed information from medical records could potentially improve research and patient care.

Electronic health records (EHRs) offer unprecedented quantities of structured and unstructured data for driving research and improving care delivery. Manually extracting relevant information from unstructured free-text EHRs is costly and laborious. Recent developments in natural language processing (NLP) and the advent of large language models (LLMs) offer promising and potentially transformational alternatives that can accurately acquire relevant information from unstructured text for patient diagnosis[1–3], as well as perform several routine tasks from medical records[4,5].

As in other medical domains, studies of antibiotic resistance, use, and stewardship have traditionally relied on manual review of clinical notes and prescriptions[6–8] or mapping of International Classification of Diseases (ICD) diagnostic codes to identify acute infection diagnoses and chronic

[1]Big Data Institute, Nuffield Department of Population Health, University of Oxford, Oxford, UK. [2]Nuffield Department of Medicine, University of Oxford, Oxford, UK. [3]University Hospitals Bristol & Weston NHS Trust, Bristol, UK. [4]NIHR Health Protection Research Unit in Healthcare Associated Infections and Antimicrobial Resistance, University of Oxford, Oxford, UK. [5]NIHR Oxford Biomedical Research Centre, Oxford, UK. [6]Institute of Biomedical Engineering, University of Oxford, Oxford, UK. [7]Oxford University Hospitals NHS Foundation Trust, Oxford, UK. [8]These authors contributed equally: Kevin Yuan, Chang Ho Yoon, Qingze Gu. ✉e-mail: kevin.yuan@ndph.ox.ac.uk

comorbidities[9,10]. However, in studies of sepsis, ICD codes identified fewer cases than clinical data[11,12], particularly in less common infections like meningitis[13], and had variable validity[14]. Additionally, since codes are frequently recorded only after patient discharge or completion of an episode of care, assigned infection sources may not align with individual antibiotic prescriptions. Conversely, manual chart review has higher sensitivity and can detect indications evolving over time, but time and cost constraints mean that case numbers are often limited.

Recent research studies have shown applying LLMs to entire medical records can effectively make predictions relevant to diagnosis, treatment and care delivery, and generate new medical content[4,5]. However, there is also a clear need for research and service applications to be able to extract specific individual features from free-text in EHRs reliably and efficiently whilst also meeting information governance requirements. These features can be extracted by scanning the whole EHR, or by targeting specific free-text forms. As an example of the latter targeted approach, we studied antibiotic prescriptions and tried to infer the type of infection or infections being treated from a free-text box completed by clinicians describing the reason (indication) for antibiotics being given.[15] We investigated several methods, comparing infections identified by clinician review of the free-text to those identified by state-of-the-art NLP models, i.e., Bidirectional Encoder Representations from Transformers (BERT)[16] and LLMs from the Generative Pre-trained Transformer (GPT) family[17], as well as classical NLP methods and regular expression-based text searches. Additionally, we also compared findings to the infections that would have been identified by traditional approaches using ICD-10 codes only.

By applying these methods, we find that modern transformer-based NLP models, such as fine-tuned Bio+Clinical BERT and GPT models, significantly outperform traditional approaches in accurately inferring infection types from free-text indications. These advanced models extract specific infection information more frequently than standard methods that rely solely on ICD-10 codes, highlighting their potential for information extraction from unstructured medical records.

## Methods

### Study design and population

We used EHRs from two distinct locations, Oxford (three hospital sites) and Banbury (one hospital), with data from Oxford serving as our training and internal test set, and Banbury as our external test set. These four hospitals collectively provide 1100 beds, serving 750,000 residents in Oxfordshire, ~1% of the UK population. Deidentified versions of this EHR data were obtained from Infections in Oxfordshire Research Database (IORD), which has approvals from the National Research Ethics Service South Central – Oxford C Research Ethics Committee (19/SC/0403), the Health Research Authority and the Confidentiality Advisory Group (19/CAG/0144) as a deidentified database that can be used for infection research without a requirement for individual patient consent. Approvals for this specific study and use of the data were granted by the IORD oversight committee, based on a written proposal. All patients aged ≥16 years who had antibiotic prescriptions and were admitted between 01-October-2014 and 30-June-2021 were included.

### 'Ground truth' labelling

Within the EHR, clinicians documented the indication for antibiotic prescriptions within a character-limited free-text field. We trained models to classify this text to identify the infection source or sources being treated, and any uncertainty expressed by clinicians. Two clinical researchers reviewed each antibiotic indication text string used for training and testing to establish a reference or 'ground truth' result for the clinical syndrome being treated (4000 unique strings for training, and 2000 randomly sampled strings the testing and external validation; see "Training, test and external evaluation" section). Inter-rater agreement was assessed using Cohen's Kappa, following which any discrepancies were resolved by a third researcher, a clinical infection specialist. Antibiotic indications were labelled using 11 categories representing infection source: Urinary; Respiratory; Abdominal;

Neurological; Skin and Soft Tissue; Ear, Nose and Throat (ENT); Orthopaedic; Other specific (i.e. another body site); Non-specific (i.e. no body site provided, e.g. "sepsis", "infection"), Prophylaxis, Not informative (i.e. text unrelated to the source of infection, e.g. "as instructed by Dr X"). Each category was recorded as a binary variable, such that more than one potential source could be recorded, e.g. the input string "urinary/chest" would be labelled as both urinary and respiratory. An additional variable was used to document the presence of uncertainty expressed by the prescriber, e.g. "urinary/chest" or "? UTI".

### Traditional classification methods

Regex rules. The most intuitive and deterministic method for classifying free-text is searching for specific keywords from a list of predefined words for a given category. We employed fuzzy regular expression (regex) matching patterns with term-specific word boundaries and variable fuzziness to allow for misspellings and variations using the regex python package (see Supplementary Methods *and* Supplementary Fig. S1).

n-grams & XGBoost. A second approach used a separate tokeniser, embedding and classifier structure; specifically, scikit-learn's n-grams & count vectorisation and the gradient boosting model architecture XGBoost[18,19]. Each free-text indication term was broken up into overlapping subwords of length $n$ and then count-vectorised, with the count representing the frequency of each subword's occurrence. The vectors of dimension *vocabulary size* were then fed as input features to the classification model. We determined the optimal n-gram size ($n$) and hyperparameters for XGBoost during model training by maximising the receiver operator curve area under the curve (ROC AUC) (details below).

### BERT Classifier

Current state-of-the-art NLP tasks employ models built on transformer architectures, with the Bidirectional Encoder Representations from Transformers (BERT) model family well suited for many tasks requiring semantic understanding. We finetuned[20] pre-trained BERT models on a single GPU instance and used BERT for both encoding and classification. We evaluated the original generic "uncased base BERT" model, pre-trained on the BooksCorpus and English Wikipedia and a domain-specific "Bio+Clinical BERT", pre-trained on biomedical and clinical text[21,22].

### Zero-shot and fine-tuned LLM classifier

Compared to BERT, the GPT family enables zero- or few-shot learning, i.e. there is potentially minimal need for labelled data for task-specific training. We developed prompts for zero-shot learning with GPT4, comprised of instructions and the target categories, asking the model to complete the categories, but without using any training data (see Supplementary Note S1 for specific prompts)[23].

We also finetuned a GPT3.5 model using the complete training dataset. Finetuning was achieved by presenting the desired output alongside the training input data. We used the same system prompt as for the GPT4 model, while providing the training examples, which were fed in batches of ten. Additional model hyperparameters, such as learning rate, and epochs, were chosen through grid search. All GPT models were run on OpenAI's platform. All data uploaded to OpenAI's services was thoroughly reviewed prior to ensuring that no personal identifiable data was included.

### Training, test and external evaluation

We divided the Oxford data with a 90/10 train/test split, resulting in a raw training set and internal test set. From the training data, we labelled and used the 4000 most frequently occurring unique indication text strings. To make labelling tractable we discarded the remaining unlabelled data from the training set. From the internal test data, we randomly selected and exhaustively labelled indications present in 2000 prescriptions. For the external test set from Banbury, we also labelled 2000 randomly selected

entries. All models were trained on the training dataset with grid-search-based hyperparameter tuning based on cross validation and tested on both the internal and external test sets.

The multi-label performance of each method was evaluated using scikit-lean's implementation of weighted F1 scores, precision-recall (PR AUC) and ROC AUC. Weighted averages take into account the varied distributions of the infection categories, such that more common categories contribute more to the overall average, producing estimates that reflect the original data source and represent real-world performance.

### Comparator classification by ICD-10 codes

We also inferred the infection being treated using only ICD-10 codes and no free-text to provide a comparison with this traditional approach. We mapped primary and secondary ICD-10 diagnosis codes from the same admission as the antibiotic prescription to the 11 infection sources using CCSR classifications[24] as an intermediate step (see Supplementary Methods and Supplementary Note S2). We then compared infection sources extracted from free-text indications to infection sources derived from ICD-10 codes.

### Reporting summary

Further information on research design is available in the Nature Portfolio Reporting Summary linked to this article.

## Results

We obtained antibiotic prescribing indication data from 826,533 prescriptions from 171,460 adult inpatients, ≥16 years, between 01-October-2014 and 30-June-2021 from three hospitals in Oxford, UK. The most commonly prescribed antibiotics were co-amoxiclav ($n = 269,945$, 33%), gentamicin ($n = 70,002$, 8%), and metronidazole ($n = 65,094$, 8%) (Supplementary Table S2), and the most common specialities were General Surgery ($n = 146,719$, 18%), Acute General Medicine ($n = 98,687$, 12%), and Trauma and Orthopaedics ($n = 90,719$, 11%) (Supplementary Table S3). Patients were a median 56 years old (IQR 36–73), and 94,721 (55%) were female (Supplementary Table S4).

We also used an independent external test dataset to assess classifier performance further, from the Horton Hospital, Banbury (~30 miles from Oxford). This dataset comprised 111,617 prescriptions from 25,924 patients between 01-December-2014 and 30-June-2021, with 13,650 unique free-text indications. Antibiotics prescribed (Supplementary Table S2) and specialities (Supplementary Table S3) were broadly similar to the Oxford training and internal test dataset. Patients were a median 67 years old (IQR 47–80), and 13,853 (53%) were female (Supplementary Table S4).

### Prescription indications

From the 826,533 Oxford prescriptions, 86,611 unique free-text indications were recorded. The top 10 accounted for 41% of all prescriptions; these

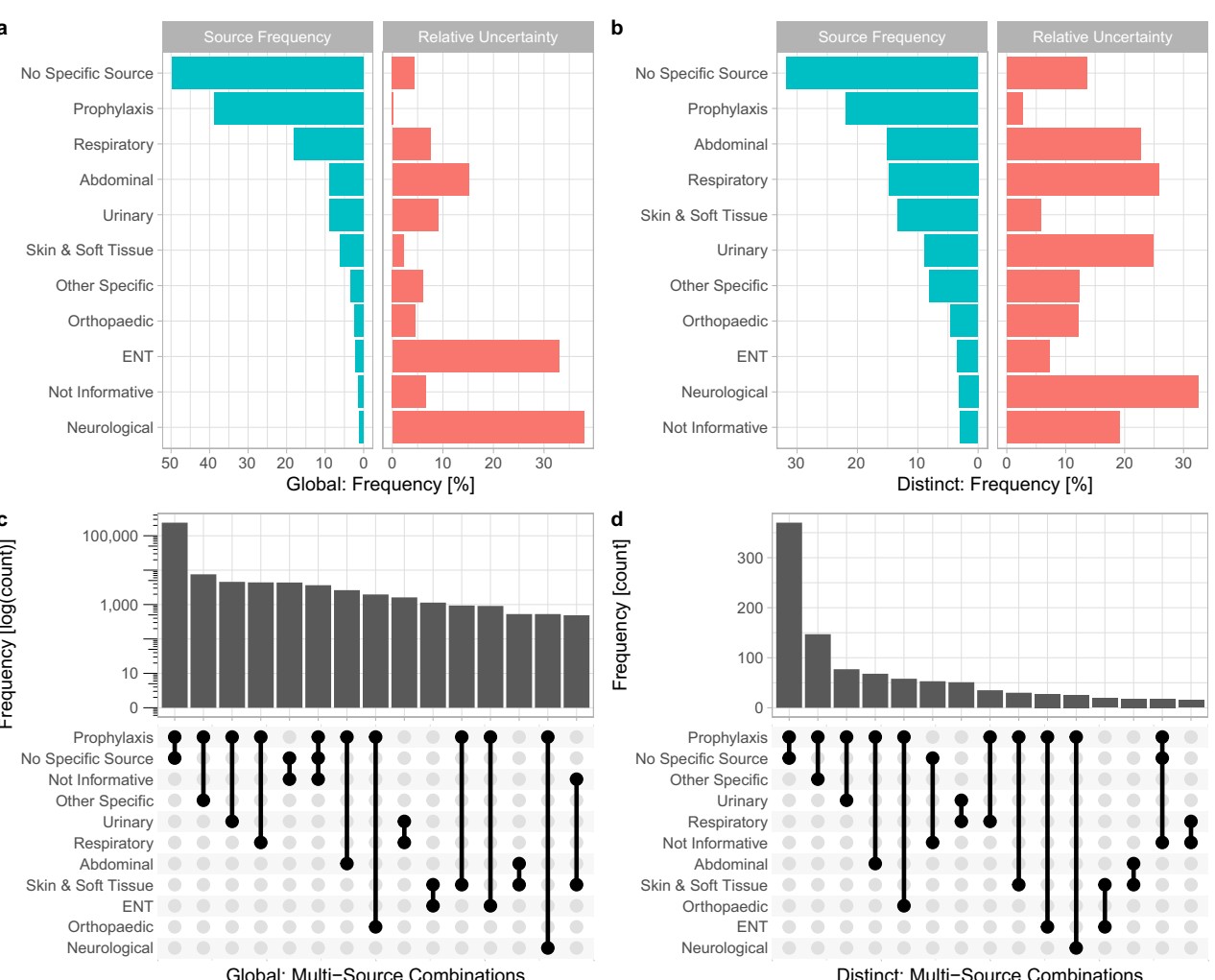

**Fig. 1 | Infection & source distributions within labelled training data from three Oxford hospitals.** Bar charts (**a**) and **b** show the distribution of the sources and the uncertainty relative to the infection source. The up-set plots (**c**) and **d** show the occurrence of multiple sources within the same prescription. **a** and **c** show distributions across the entire labelled indications training set ("global"), **b** and **d** across a distinct set of 4000 most common indications ("distinct"). *Indications falling into ENT such as "neck abscess" were often also labelled with Skin & Soft Tissue.

included "Perioperative Prophylaxis" (20%), "UTI" (4%), "LRTI" (3%), "Sepsis" (3%), and "CAP" (3%). The most commonly occurring 4000 unique indications, used for model training, accounted for 84% (692,310) of prescriptions (Supplementary Fig. S2 *and* Table S5).

As expected, different wording was used to reflect similar concepts, e.g. "CAP [community acquired pneumonia]", "LRTI [lower respiratory tract infection]", "chest infection", and "pneumonia". Additionally, misspellings were common, e.g. "infction" [infection], "c. dififcile" [*C. difficile*]. Multiple examples expressed uncertainty, or multiple potential sources of infection, e.g. "sepsis?source", "UTI/Chest", etc. Reflecting the complexity of prescribing, there were multiple potentially informative, but rarely occurring indications, e.g., "transplant pyelonephritis", "Ludwig's angina", and "deep neck infection", which were only seen 51 (<1%), 27 (<1%), and 13 (<1%) times respectively.

### 'Ground truth' labels

Following labelling by clinical experts, the 4000 most commonly occurring free-text indications were classified into 11 categories, with a separate variable capturing the presence of uncertainty. There was generally close agreement between the two initial clinical researchers classifying the text strings (average Cohen's Kappa = 0.80, range across categories 0.55–0.96, Supplementary Table S6). The most commonly assigned sources were "Prophylaxis" (267,788/692,310 prescriptions, 39%), "Respiratory" (125,744, 18%) and "Abdominal" (61,670, 9%). 50% (*n* = 344,773) prescriptions had "No Specific Source". The most uncertainty was expressed in "Neurological" and ENT cases at 38% and 33%, respectively (Fig. 1a).

Although "Respiratory" was the most common category overall after "Prophylaxis", there were more distinct text strings associated with "Abdominal" infections, with "Skin and Soft Tissue" infection also having a disproportionately larger number of unique text strings (Fig. 1b). Most 'multi-source' prescriptions were a combination of "Prophylaxis" and a source (>90%). Excluding prophylaxis, the most common combinations of sources were "No Specific source" and "Not Informative", "Urinary" and "Respiratory", and "Skin & Soft Tissue" and ENT, in 1.6%, 0.58%, 0.41% prescriptions, respectively (Fig. 1c, d). The former two reflected diagnostic uncertainty and the latter reflected infections of the face, head and neck frequently involving skin/soft tissue.

### Classifier performance

We trained classifiers using the labelled training data from three Oxford hospitals (Fig. 2). Compared to clinician-assigned labels, within the internal Oxford test dataset (*n* = 2000), the weight-averaged F1 score across classes was highest using Bio+Clinical BERT (Average F1 = 0.97 [worst performing category F1 = 0.84, best performing F1 = 0.98]) followed by fine-tuned GPT3.5 (F1 = 0.95 [0.77–0.99]), base BERT (F1 = 0.93 [0.23–0.98]) and tokenisation+XGBoost (F1 = 0.86 [0.64–0.96]). Nearly all approaches exceeded traditional regular expression-based matching (F1 = 0.71 [0.00–0.93]). The zero-shot GPT4 model, which did not require labelled data, performed similarly to this baseline (F1 = 0.71 [0.30–0.98]) (Table 1, additionally shows classification run times and 95% confidence intervals). Similar performance characteristics were achieved on the external validation dataset from Banbury (*n* = 2000; weight-averaged F1 scores: Bio+Clinical

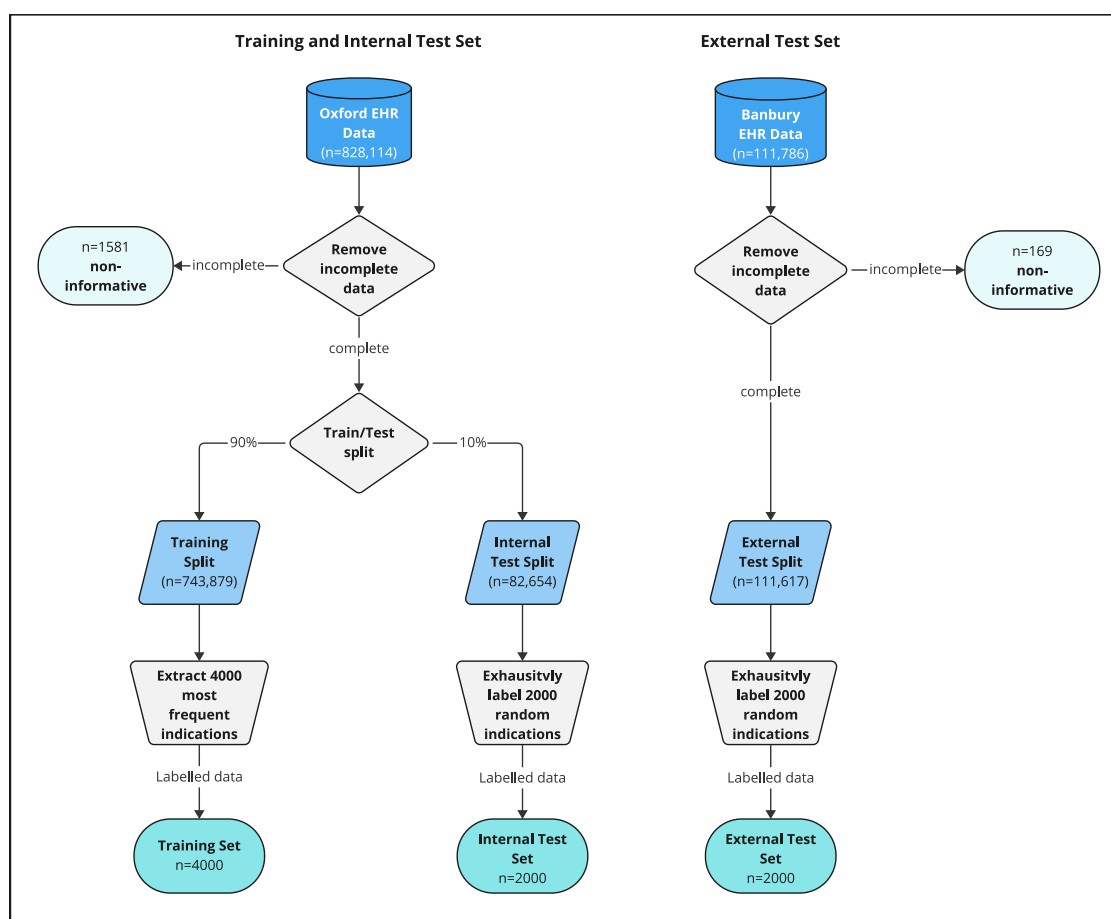

**Fig. 2 | Data processing flow chart for training and internal and external test datasets.** Prescribing data was fetched from EHR databases and filtered for complete data, where incomplete data refers to entries without usable indications (e.g. blank, single character punctuation [excluding a question mark] or "NA"). The 4000 most frequent indications within the training split were labelled, all remaining training data was discarded. 2000 entries were randomly sampled from both the internal and external test datasets and exhaustively labelled, resulting in a total of three datasets (training set, internal test set, external test set).

**Table 1 | Model performance metrics for the internal (Oxford) test set**

| Model | F1 Score | | | ROC AUC | | | PR AUC | | | Per-Category Accuracy | | | Accuracy | Training Runtime | Test Runtime |
|---|---|---|---|---|---|---|---|---|---|---|---|---|---|---|---|
| Aggregation | Average [95% CI] | Low | High | Average [95% CI] | Low | High | Average [95% CI] | Low | High | Average [95% CI] | Low | High | Overall [95% CI] | Per 4k | Per 10k |
| Regex | 0.71 [0.70, 0.73] | 0.00 | 0.93 | - | - | - | - | - | - | 0.82 [0.81, 0.82] | 0.32 | 0.99 | 0.14 [0.13, 0.16] | - | 6.4 s |
| XGBoost | 0.86 [0.85, 0.87] | 0.64 | 0.96 | 0.96 [0.96, 0.97] | 0.87 | 0.99 | 0.90 [0.88, 0.91] | 0.62 | 0.99 | 0.95 [0.95, 0.96] | 0.92 | **1.00** | 0.72 [0.70, 0.74] | 6 s | **1.2 s** |
| Base BERT | 0.93 [0.92, 0.94] | 0.23 | 0.98 | 0.99 [0.98, 0.99] | 0.91 | 0.99 | 0.97 [0.96, 0.98] | 0.69 | 0.99 | 0.98 [0.98, 0.99] | 0.97 | 0.99 | 0.88 [0.87, 0.90] | 282 s[1] | 82.2 s[1] |
| **Bio+Clinical BERT** | **0.97 [0.96, 0.97]** | **0.84** | 0.98 | **0.99 [0.99, 1.00]** | 0.96 | **1.00** | **0.98 [0.98, 0.99]** | 0.88 | **1.00** | **0.99 [0.99, 0.99]** | **0.98** | **1.00** | **0.94 [0.92, 0.95]** | 279 s[1] | 83.1 s[1] |
| Fine-Tuned OpenAI GPT3.5 | 0.95 [0.94, 0.96] | 0.77 | **0.99** | - | - | - | - | - | - | 0.98 [0.98, 0.98] | 0.97 | **1.00** | 0.91 [0.90, 0.92] | ~3500 s[2] | ~3000 s[2] |
| Zero-Shot OpenAI GPT4 | 0.71 [0.69, 0.72] | 0.30 | 0.98 | - | - | - | - | - | - | 0.87 [0.86, 0.88] | 0.64 | **1.00** | 0.50 [0.48, 0.52] | - | ~3000 s[2] |

Each score is listed with the weighted average across the classes (sources), with the lowest and highest performing class. Overall Accuracy refers to the score calculated for a sample treated as a whole. The best scores for each metric are highlighted in bold. The fine-tuned Bio+Clinical BERT outperforms all other methods on both internal and external test sets. The 95% confidence intervals (CI) shown were calculated using 1000 bootstrap iterations.
[1]Using one Nvidia V100 GPU.
[2]OpenAI's cloud service.

BERT 0.98 [0.87–1.00], fine-tuned GPT3.5 0.97 [0.70–1.00], Base BERT 0.97 [0.63–0.99], XGBoost 0.84 [0.63–1.00], Regex 0.74 [0.00–0.96], GTP4 0.86 [0.25–1.00]) (Table 2).

### Classifier performance by class
Using the best-performing classifier, Bio+Clinical BERT, we assessed performance within each category. The best-performing categories within our internal test set were "Respiratory", "No Specific Source" and "Prophylaxis" (F1 Score=0.98), followed by "Urinary" (0.97), "Abdominal" (0.96), "Orthopaedic" (0.90), "Not Informative" (0.89) and "Neurological" (0.88). The worst performing category was Orthopaedic (0.84), likely due to the high variety of terms used and low number of training samples ($n = 14$, Supplementary Table S7). Uncertainty was also well detected (0.96) (Fig. 3a, Supplementary Table S8).

In the external test data, scores varied slightly, with all source categories except for "Not Informative" having F1 scores on average 0.02 higher compared to the internal test set. These small differences likely arose from different compositions of categories and the amount of shared vocabulary between the training and test datasets.

### Misclassifications
Most misclassifications were spread evenly across classes for single indications. The two most common misclassifications occurred for "Orthopaedic" and "Other Specific" cases, with 12% being misclassified as "Prophylaxis" and 8% as "Skin and Soft Tissue", respectively, on the internal test set. On the external test set, most misclassifications were predicted to be "Other Specific" or "Prophylaxis" (Fig. 3c).

### Training dataset size
We examined the effect of training size on model performance using randomly selected training dataset subsets of 250, 500, 750, 1000, 1500, 2000, 3000, and 4000 unique indications, tested using both internal/external test sets. There was a notable increase in performance (AUC-ROC and F1 scores) when the training size increased from 250 to 1000 samples, suggesting a minimum of 1000 training samples for adequate performance. However, we saw limited improvement as the training dataset size rose to 4000, indicating there may be only marginal gains to expanding the training data beyond 4000 samples (Supplementary Fig. S3).

### Comparing free-text indications to ICD-10 codes
We also compared infection sources from manually labelled 'ground truth' free-text indications to sources inferred from ICD-10 diagnostic codes. 31% of sources classified as "unspecific" using diagnostic codes could be resolved into specific sources using free-text. Rarer infection sources such as "CNS" and "ENT" (<1% and no occurrence in diagnostic codes) were represented better by sources extracted from free-text (4% and 4% respectively). Overall, where defined, sources listed in clinical codes generally concurred with the 'ground truth' free-text sources (Fig. 4).

### Discussion
We show that modern NLP methods can extract clinically relevant details from semi-structured free-text fields. In our example application, a fine-tuned Bio+Clincal BERT transformer model classified the infection source being treated using clinician-written antibiotic indication text with an F1 score of 0.97 (harmonic mean of sensitivity and positive predictive value). Although this required manual labelling of several thousand text strings, exhaustive labelling of all possible prior strings was not required to achieve this performance. The overall accuracy of the Bio+Clinical BERT model was substantially better than a sophisticated regular expression-based approach (accuracy 0.94 vs. 0.14), despite the latter being the solution that many healthcare institutions and researchers might have previously chosen due to ease of implementation.

We also explored what performance might be possible using LLMs without having to label data, e.g. where this is not possible or too resource intensive. However, zero-shot learning with GPT4 only achieved modest

## Table 2 | Model performance metrics for the external (Banbury) test set

| Model | F1 Score | | | ROC AUC | | | PR AUC | | | Per-Category Accuracy | | | Accuracy |
|---|---|---|---|---|---|---|---|---|---|---|---|---|---|
| *Aggregation* | *Average [95% CI]* | *Low* | *High* | *Average [95% CI]* | *Low* | *High* | *Average [95% CI]* | *Low* | *High* | *Average [95% CI]* | *Low* | *High* | *Overall [95% CI]* |
| *Regex* | 0.74 [0.73, 0.75] | 0.00 | 0.96 | - | - | - | - | - | - | 0.82 [0.82, 0.83] | 0.41 | 0.99 | 0.24 [0.22, 0.26] |
| *XGBoost* | 0.84 [0.83, 0.85] | 0.63 | **1.00** | 0.94 [0.93, 0.94] | 0.86 | **1.00** | 0.87 [0.85, 0.88] | 0.57 | **1.00** | 0.94 [0.93, 0.94] | 0.88 | **1.00** | 0.68 [0.66, 0.70] |
| *Base BERT* | 0.97 [0.96, 0.98] | 0.63 | 0.99 | **0.99 [0.99, 1.00]** | 0.95 | **1.00** | **0.98 [0.98, 0.99]** | 0.75 | **1.00** | **0.99 [0.99, 1.00]** | **0.99** | **1.00** | 0.95 [0.94, 0.96] |
| ***Bio +Clinical BERT*** | **0.98 [0.98, 0.99]** | **0.87** | **1.00** | **0.99 [0.99, 1.00]** | 0.97 | **1.00** | **0.98 [0.98, 0.99]** | **0.87** | **1.00** | 0.99 [0.99, 1.00] | **0.99** | **1.00** | **0.97 [0.96, 0.97]** |
| *Fine-Tuned OpenAI GPT3.5* | 0.97 [0.96, 0.97] | 0.70 | **1.00** | - | - | - | - | - | - | **0.99 [0.99, 0.99]** | 0.98 | **1.00** | 0.95 [0.94, 0.96] |
| *Zero-Shot OpenAI GPT4* | 0.86 [0.85, 0.87] | 0.25 | **1.00** | - | - | - | - | - | - | 0.95 [0.94, 0.95] | 0.81 | **1.00** | 0.73 [0.71, 0.75] |

Each score is listed with the weighted average across the classes (sources), with the lowest and highest performing class. Overall Accuracy refers to the score calculated for a sample treated as a whole. The best scores for each metric are highlighted in bold. The fine-tuned Bio+Clinical BERT outperforms all other methods on both internal and external test sets. The 95% confidence intervals (CI) shown were calculated using 1000 bootstrap iterations.

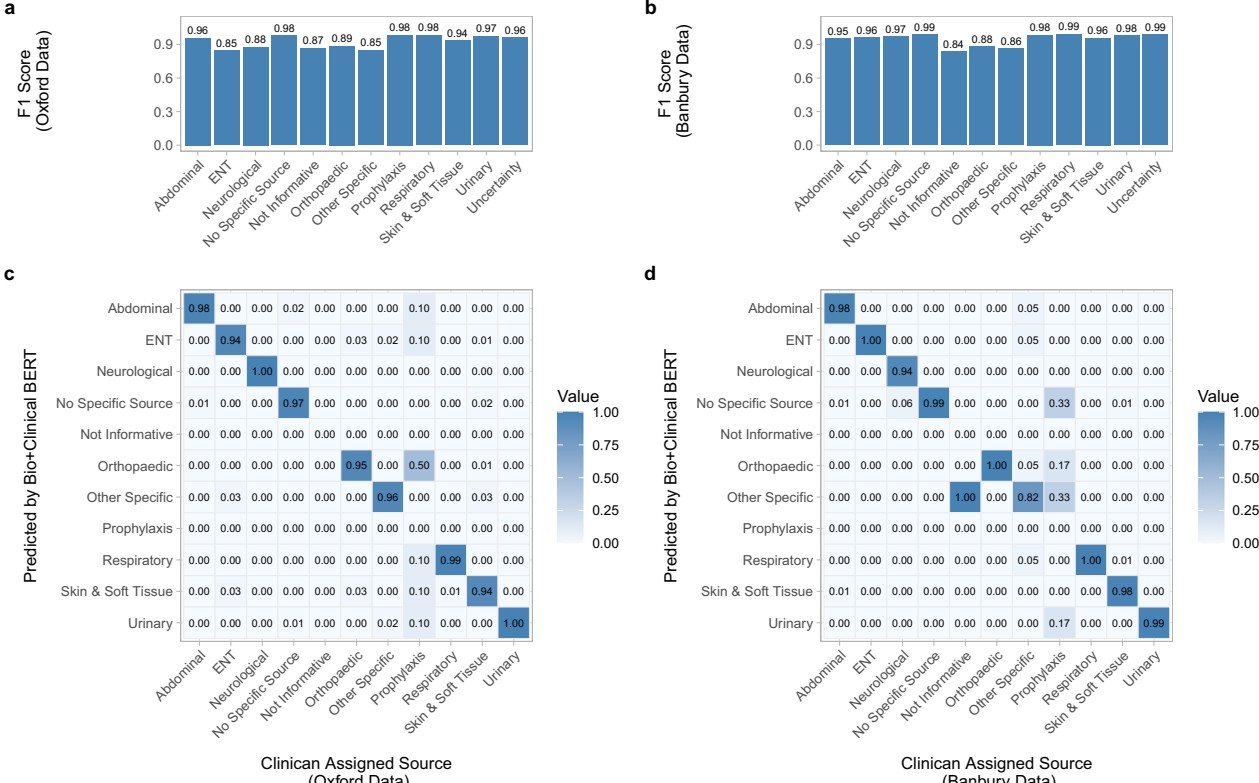

**Fig. 3 | Performance metrics for Bio+Clinical BERT on both internal and external test sets.** Bar charts (**a**) and **b** show the per-category prediction performance. Confusion matrices (**c**) and **d** are single indication test prescriptions and show model prediction errors across the sources for given ground truths (clinician-assigned sources). **a** and **c** show evaluations performed on the internal test set from three Oxford hospitals, **b** and **d** on the external test set from the Banbury hospital.

performance, but it was still similar to the baseline regex method and more consistent across classes. Using LLMs with labelled training data, i.e. a fine-tuned GPT3.5 variant, achieved results comparable to the Bio+Clinical BERT approach when correctly specified and tuned but could be more challenging to deploy as responses can vary in formatting, making it difficult to parse correctly into a rigid format required for most downstream tasks or EHR systems. In environments with limited computing resources where the deployment of deep-learning models is not feasible, regex and XGBoost-based models provide possible alternatives with a reduced runtime of 6.4 and 1.2 s/10k indications vs. 82.2 s/10k for Bio+Clinial BERT.

**Fig. 4 | Comparing infection sources between clinician-assigned free-text indications (left) and diagnostic codes (right) in the training and internal and external test sets.** Clinician-assigned categories were extracted from prescribing data and manually labelled, diagnostic codes sources calculated from procedure and discharge codes.

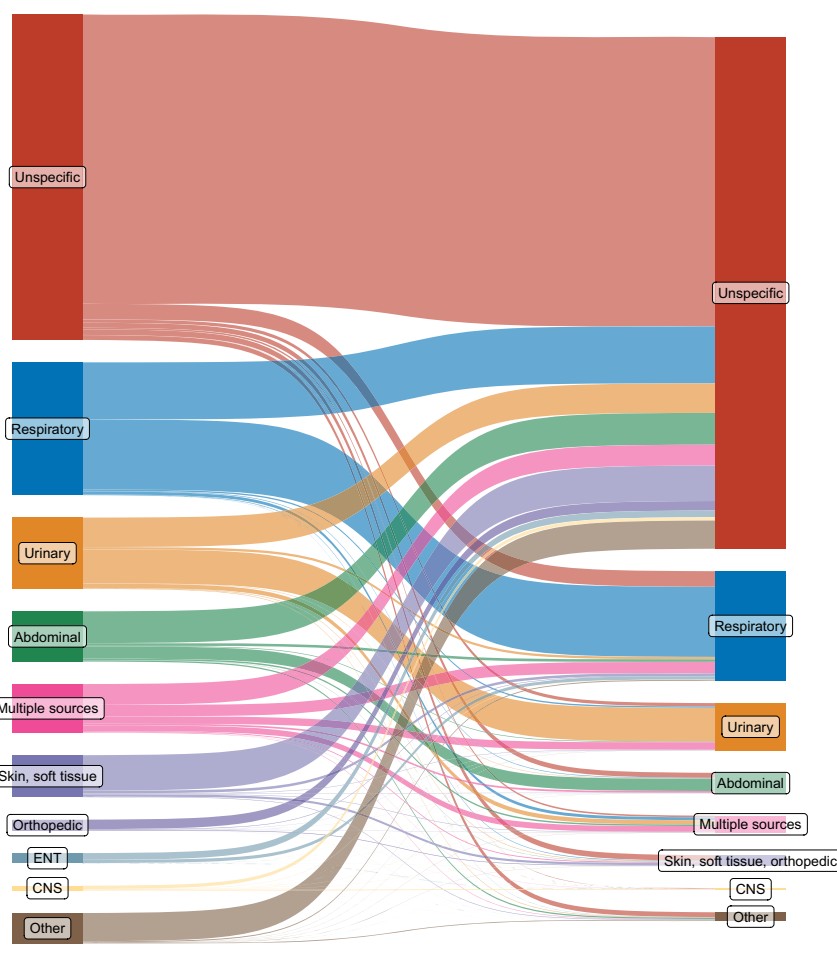

Currently, research or clinical use of free-text may be limited by concerns that personal data may be included. Here, by homogenising and categorising sensitive free-text data and using a locally run BERT model, we present a privacy-aware solution that enables researchers to utilise the depth of free-text data without direct access or the possibility of identifying specific patients.

Our study has several limitations, including that we only used a subset of the available training data, through the non-exhaustive labelling of a subset of antibiotic indication text strings. However, labelling the 4000 most common unique terms, accounting for 84% of the data, achieved very high performance, with sensitivity analyses suggesting that labelling more examples would not have improved performance substantially. This is likely possible because the underlying Bio+Clinical BERT model is already pretrained on medical terms and capable of inferring similar words, with the nature of the data suggesting that there is a relatively finite number of unique terms (excluding misspellings). Of note, many of the remaining 17% of text strings were different combinations of already labelled words, suggesting fewer "new" or unseen keywords than might be expected. Not fully labelling the training data also makes it more difficult to compare category distributions with the test datasets. We also only used a subset of the test data to evaluate performance; however, the 2000 randomly selected samples are likely representative. Although the labelling process was somewhat subjective, independent labelling by two clinical researchers was largely consistent (average Cohen's Kappa 0.80), with a third clinical researcher adjudicating any discrepancies able to minimise this.

Future enhancements could include smaller, more efficient NLP models that might better balance computational demands and performance. Techniques such as model pruning, quantisation, and knowledge distillation could reduce model size and computational requirements while preserving performance[25–27]. While GPT4 deployments can comply with data governance requirements, its use presents challenges in some settings, as it is usually accessed via third-party cloud compute providers rather than healthcare institutions. Where data need to remain on site, open-source, locally-deployed language models, such as LLAMA, ALPACA or Mistral 7B, may be alternatives that could be further investigated[28–30].

Our approach has several possible applications; for example, it could be used to monitor and evaluate prescribing practice across different conditions, it provides classification of possible infection sources for epidemiological research[31], and is also a mechanism for extracting standardised features from medical records for use in predictive algorithms being developed to improve patient care. Although we demonstrate excellent performance for antibiotic indications, it could also be applied to other short strings of free-text, for example descriptions of surgical procedures, patient functional states, or presenting complaints in emergency department and hospital admission data.

In summary, we show that state-of-the-art NLP can be used to efficiently and accurately categorise semi-structured free-text in medical records. This has the potential to be applied widely to analyse medical records more accurately and at scale, potentially opening opportunities for

**Article**

new epidemiological and intervention studies across medicine, as well as possibilities for improving care delivery.

## Data availability

The data analysed are available from the Infections in Oxfordshire Research Database (https://oxfordbrc.nihr.ac.uk/research-themes/modernising-medical-microbiology-and-big-infection-diagnostics/infections-in-oxfordshire-research-database-iord/), subject to an application and research proposal meeting on the ethical and governance requirements of the Database. Labelled training and test datasets and the pre-trained BERT model are also available via an application to the Database. The source data for Fig. 1 is in Supplementary Data 1. The source data for Fig. 2 is in Supplementary Data 2. The source data for Fig. 3 is in Supplementary Data 3. The source data for Fig. 4 is in Supplementary Data 4.

## Code availability

All tools developed for this study (Regex builder, Bio+Clinical BERT pipeline, GPT3.5 finetuning and GPT4 zero-shot request tools) and the evaluation frameworks are available through GitHub and Zenodo[32]: https://github.com/kevihiiin/EHR-Indication-Processing/ https://doi.org/10.5281/zenodo.13987740.

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

## Acknowledgements

This work was supported by the National Institute for Health Research Health Protection Research Unit (NIHR HPRU) in Healthcare Associated Infections and Antimicrobial Resistance at Oxford University in partnership with the UK Health Security Agency (NIHR200915), and the NIHR Biomedical Research Centre, Oxford. D.W.E. is a Big Data Institute Robertson Fellow. A.S.W. is an NIHR Senior Investigator. The views expressed are those of the authors and not necessarily those of the NHS, the

NIHR, the Department of Health or the UK Health Security Agency. K.Y. is supported by the EPSRC Centre for Doctoral Training in Health Data Science (EP/S02428X/1). The funders had no role in study design, data collection and analysis, decision to publish, or preparation of the manuscript.

## Author contributions

D.W.E., T.Z., A.S.W., K.Y. and C.H.Y. conceived the study. K.Y., C.H.Y., Q.G., H.M. and D.W.E. analysed the data. K.Y. and D.W.E. wrote the first draft of the manuscript. All authors contributed to revising the manuscript.

## Competing interests

The authors declare no competing interests.
