## [Transparent Peer Review file · Communications Medicine]

Transformers and large language models are efficient feature extractors for electronic health record studies

Corresponding Author: Mr Kevin Yuan

Version 0:

Reviewer comments:

Reviewer #1

(Remarks to the Author)

This paper compared fine-tuned domain-specific Bio+Clinical BERT model with traditional regex and n-grams/XGBoost. They also used a few-shot OpenAI GPT4 model and a fine-tuned GPT3.5 model. The task was to extract features from hospital antibiotic prescriptions. They found that fine-tuned BERT-based models outperformed LLMs for structured tasks, and few shot LLMs match the performance of traditional NLP without the need for labelling. This is a novel finding, and highlights that state-of-the-art NLP can be used to categorise semi-structured free-text in medical records.

The paper is well-structured and clearly written, making it easy to read and understand.

A few minor comments:

General comment: I have generally seen ICD-10 written with a hyphen while the manuscript writes it as ICD10 without the hyphen.

Line 65: Since GPT is an LLM, this can be rephrased as LLMs such as GPT instead of separating them out.

Line 72 mentions getting data from two hospital sites, and lines 141 and 180 state getting data from three hospitals - might help to clarify if it is two or three?

Line 145-146 - is information on ethnicity available? It would be good to state that as well, along with the age and gender frequencies.

Line 165 - Ground truth labels subsection - are you able to provide inter-annotator agreement scores for the two clinical researchers who labelled the data? This might shed some light on why some labels performed better than others after training.

Line 309, 498 - the github link needs updating. It seems to be broken.

Line 422 - Table 1 - would it be possible to add 95% confidence intervals to the F1-scores? This will provide a better picture of actual differences between the models. A common method used to calculate 95% confidence intervals is bootstrapping.

Privacy comment: Were you able to download local versions of the GPT models to use within the secure hospital framework to conduct this work? I noticed the footnote under line 429 says OpenAI's cloud service. Was the data shared with OpenAI? Some clarity on this would be great!

Reviewer #2

(Remarks to the Author)

Review

The Manuscript by Yuan et al. titled "Transformers and large language models are efficient feature 1 extractors for electronic health record studies" describes comparison of various natural language processing methods in the analysis of EMR data using antibiotic prescriptions as model. With the advances in AI and machine learning such study is very useful since the need for fast robust and user-friendly methods which would allow users to analyze a vast ocean of data is very high currently. The study is well designed, the analysis seems properly performed with valid conclusion and insightful discussion. The only minor problem is the presentation which is a bit haphazard and would benefit from some correction – listed below.

1. Lines 51-53 – should be re-phrased since is not clear at all what authors want to say here
2. L. 53 – 'codes recorded only after discharge' – is not correct actually, and if this remains a statement, then appropriate reference is required.
3. L. 78 – need to indicate here the time period which covered the extracted EMR data
4. L. 80-81 – what exactly were clinicians reviewing – the random subset of 4000 and 2000 text string, or all text strings – this is not clear at all.
5. Overall, a better schematic of the study should be presented, with a glossary of what the authors mean by each term (indication, labelling, category, infections source, class, etc.) – otherwise it is not clear whether class and category is the same thing or not.
6. When describing technical details, need to reference more of something is not explained in detail.
7. Which software was used to calculate statistical parameters like ROC-AUS, F1 scores etc.?
8. LL. 135-138 are not clear – why multi-label performance (what do authors mean by it exactly), what varied distribution – 'classes' appear here for the first time, and then again in L. 182, is it the same as 'category'?
9. L. 157 – reference to Appendix S7 – Labelling coverage – needs clarification – what is the line that goes beyond the 4000 set size – if only 4000 were labeled – is it prediction? Doesn't look like it. And what is 'sample' in the figure legend. Please, revise to make it clear.
10. Also, it would be good to have more examples of various text-strings as a supplemental file.
11. Figure 1 needs better Legend with exact description of what authors mean by 'global' and 'distinct'; also, panels A and B would look better if they had categories in same order – easier to compare side-by-side.
12. Figure 2 and Methods/Results – 'incomplete' – example what is meant by it?
13. When comparing BERT model's performance to other models, authors use word 'substantially better' (L. 231) – is there a way to measure whether this performance is different significantly?
14. AL. 232 – some comment or reference regarding the statement that regex 'might be chosen by many health institutions at present' would be nice.

Reviewer #3

(Remarks to the Author)

The paper focuses on automated classification and structuring of the information in clinical free text reports, with a focus on infection. From my understanding, the classification task focuses primarily on classifying individual infection indication terms that occur in infection reports. The study mainly focuses on comparing a set of machine learning models for this classification task.

As a reviewer, my expertise is mainly in machine learning, so it is difficult for me to say too much about the clinical novelty and relevancy of this work. From a machine learning perspective, there is no novelty in the methods used. Still, the comparison of classification methods/models for this task could be of some value to the health informatics community. Currently this paper lacks clarity in its explanations of how the data is prepared and how the classification task is defined and conducted. There are also some seemingly odd choices to be found regarding how the classification models are used, especially when it comes to the BERT model. Finally, although I might be mistaken and, the task seems to be relatively easy from a machine learning perspective. The variation to expect in the set of units to be classified (individual infection terms) is likely somewhat limited for many of the classes, this is also one reason why the performance scores are relatively high (close to 1 for several performance metrics).

When I started to read the paper, I was quite sure you were focusing on several classification tasks, such as prescribed drugs, prescribing specialties, and infection indications/syndrome. For a time I was also certain that you were performing some kind of named entity recognition (NER, the task of identifying and labeling word sequences within sentences). However, when looking at the results, it seems that this is not the case. I suggest that you go through the paper and try to be more consistent in how you describe the task in the different sections.

"... each antibiotic indication text string ..." – it is currently not fully clear to me what an antibiotic indication text string/term is. Please describe in detail how these are defined, identified and extracted for the manual labeling and later for the automated classification. Some of your examples seem to be "urinary/chest" and "sepsis ?source".

Continuing the above – Is it so that each of these indicators are classified individually and out of context? This needs to be specified in more detail. I think a figure (occurring early in the main article) with a clear illustrative example based on a single (anonymized or made up) report would be very helpful for enable the reader to understand the classification task(s?), the units of classification, and how these are labeled.

The task, as I understand it so far, is to classify the individual indication term(?). If my understanding is correct, I am tempted to believe that this set of unique terms is actually more or less finite (+ typos), at least for several of the classes, and within

the same unit. This also explains why the scores are close to 1 for the different metrics for the best performing methods/models. It would be nice to see a discussion around the nature of the data and the task in the paper. I assume you are also interested in infection detection on the level of individual patients. Do you think the current performance scores looks the same on that level?

“An additional variable was used to document the presence of uncertainty expressed by the prescriber” – This requires more explanation. Are these somehow used in the classification task(s)? Consider also incorporating this in the requested example figure.

“Comparator Classification by ICD10 Codes” – It is not clear what these are used for. Are these used to generate labels for an additional classification task perhaps? Or maybe some additional reference data? If not, what are these used for? And, can you assume that these are available at classification time? Please clarify.

“(details below)” – Please replace these with more specific pointers.

“Few-Shot and finetuned LLM Classifier” – Please provide more details how generative LLMs are used to classify a report. Why not show how the output looks like in Appendix S3?

Continuing the above - Did you perhaps encounter any problems of unexpected outputs from these? If so, how where these handled?

“BERT Classifier” – It is a bit unclear how the data is presented to the model and how you performed the classifications. Are you perhaps applying the classification layer to the [CLS] token? If so, does this mean you give the model one indication string at a time to classify? BERT excels at contextualized classifications; thus, this sounds like an unused potential. Why not give the full report to the model and perform multi-label classification? You may or may not want to first remove any ‘non-indication terms’. If you need to classify the individual tokens, one option would be to classify each token in a NER-CRF based manner with BERT (e.g., see TFAutoModelForTokenClassification in the Huggingface library).

Table 1 – I suggest that you report both Micro and Macro scores for the various metrics. This will provide a better insight into the classification performances (which may or may not be influenced by label imbalances).

Version 1:

Reviewer comments:

Reviewer #1

(Remarks to the Author)

The authors have addressed all my comments and concerns, I have no further comments.

Reviewer #2

(Remarks to the Author)

I am satisfied with authors' corrections and do not have any more comments.

Reviewer #3

(Remarks to the Author)

Thank you for the replies to my comments and questions. Most of my questions and concerns have been addressed. It is now quite clear how the task is formulated and what the unit of classification is, and the experiments and evaluations look good. Here are my comments:

Could you clarify whether or not you performed some text preprocessing of the indication texts? You mention at least some filtering being done the caption of Figure 2. E.g., did you normalize the text in some way, remove some special characters, and/or did you perhaps apply some (sentence) splitting to potentially split the original indication text (from the free-text box) into multiple indication text units to be classified? If any such preprocessing was used, I think it would be good to clarify this in the manuscript.

In Table S8.2 - You probably meant "Common" instead of "Uncommon".

I am somewhat wondering if the term/phrase "feature extractors" ("feature extraction") is really representative for the task and experiments that you report here. I guess it is ok, but you could give it some thought.

Finally, as mentioned in the previous review round, I think that the comparison of these methods at this classification task is valuable and interesting.

13th September 2024

Dear Reviewers,

Re: COMMSMED-24-0650

Thank you for your feedback and the opportunity to respond to your suggestions, which we do below.

Best wishes,

Kevin Yuan and David Eyre on behalf of all authors

Reviewer #1 (Remarks to the Author):

This paper compared fine-tuned domain-specific Bio+Clinical BERT model with traditional regex and n-grams/XGBoost. They also used a few-shot OpenAI GPT4 model and a fine-tuned GPT3.5 model. The task was to extract features from hospital antibiotic prescriptions. They found that fine-tuned BERT-based models outperformed LLMs for structured tasks, and few shot LLMs match the performance of traditional NLP without the need for labelling. This is a novel finding, and highlights that state-of-the-art NLP can be used to categorise semi-structured free-text in medical records.

The paper is well-structured and clearly written, making it easy to read and understand.

A few minor comments:

General comment: I have generally seen ICD-10 written with a hyphen while the manuscript writes it as ICD10 without the hyphen.

We have changed ICD10 to ICD-10 throughout.

Line 65: Since GPT is an LLM, this can be rephrased as LLMs such as GPT instead of separating them out.

We have rephrased this as suggested.

Line 72 mentions getting data from two hospital sites, and lines 141 and 180 state getting data from three hospitals - might help to clarify if it is two or three?

Data came from four hospitals, three in Oxford and a fourth at second site in Banbury. We have clarified this in the methods to reflect the two locations and four sites.

Line 145-146 - is information on ethnicity available? It would be good to state that as well, along with the age and gender frequencies.

We have added information on ethnicity to the supplement, our population is predominantly of white ethnicity.

Line 165 - Ground truth labels subsection - are you able to provide inter-annotator agreement scores for the two clinical researchers who labelled the data? This might shed some light on why some labels performed better than others after training.

We have calculated Cohen's Kappa Scores to determine the inter-annotator agreement across the individual classes and overall. The scores are now referenced in the manuscript with a full table included in the appendix.

Line 309, 498 - the github link needs updating. It seems to be broken.

We have made the Github repository public, the link should work properly now.

Line 422 - Table 1 - would it be possible to add 95% confidence intervals to the F1-scores? This will provide a better picture of actual differences between the models. A common method used to calculate 95% confidence intervals is bootstrapping.

We added 95% CI intervals to all averaged scores in Table 1.

Privacy comment: Were you able to download local versions of the GPT models to use within the secure hospital framework to conduct this work? I noticed the footnote under line 429 says OpenAI's cloud service. Was the data shared with OpenAI? Some clarity on this would be great!

Unfortunately, the model weights for the GPT3.5 and GPT4 analyses were not available to download and local deploy. We carefully reviewed all data uploaded to OpenAI to ensure that no personal identifiable data was included.

A comment has been added to the LLM classifier section in the manuscript, addressing the privacy comment.

Reviewer #2 (Remarks to the Author):

Review

The Manuscript by Yuan et al. titled "Transformers and large language models are efficient feature 1 extractors for electronic health record studies"

describes comparison of various natural language processing methods in the analysis of EMR data using antibiotic prescriptions as model. With the advances in AI and machine learning such study is very useful since the need for fast robust and user-friendly methods which would allow users to analyze a vast ocean of data is very high currently. The study is well designed, the analysis seems properly performed with valid conclusion and insightful discussion. The only minor problem is the presentation which is a bit haphazard and would benefit from some correction – listed below.

1. Lines 51-53 – should be re-phrased since is not clear at all what authors want to say here

We rephrased the sentence to express that using ICD-10 codes has been shown to be less sensitive for detecting sepsis cases compared to review of the underlying medical notes and clinical data.

2. L. 53 – ‘codes recorded only after discharge’ – is not correct actually, and if this remains a statement, then appropriate reference is required.

We have qualified the statement and changed the text into “frequently updated” to reflect that it is common practice in our own hospitals and other UK hospital trusts.

3. L. 78 – need to indicate here the time period which covered the extracted EMR data

We previously provided this at the start of the results. However, we now also added the time period from which the data were extracted (01-October-2014 and 30-June-2021) to the Methods text as well.

4. L. 80-81 – what exactly were clinicians reviewing – the random subset of 4000 and 2000 text string, or all text strings – this is not clear at all.

The clinicians reviewed the training and testing data used for the model, consisting of the 4000 most common text strings and all 2000 randomly sampled strings for the testing set. We have clarified this in the text.

5. Overall, a better schematic of the study should be presented, with a glossary of what the authors mean by each term (indication, labelling, category, infections source, class, etc.) – otherwise it is not clear whether class and category is the same thing or not.

We have added a glossary of used terms to our appendix. We have also avoided using both class and category to describe similar concepts, now using category throughout.

6. When describing technical details, need to reference more of something is not explained in detail.

We have added further clarifications as requested by all three reviewers and carefully reviewed the manuscript in full. If there are outstanding suggestions, we would be happy to incorporate these into our manuscript.

7. Which software was used to calculate statistical parameters like ROC-AUS, F1 scores etc.?

We have updated the manuscript to include that we used scikit-learn's implementation for the evaluation scores.

8. LL. 135-138 are not clear – why multi-label performance (what do authors mean by it exactly), what varied distribution – ‘classes’ appear here for the first time, and then again in L. 182, is it the same as ‘category’?

Each substring is classified across multiple categories. For example, “?Chest/UTI” is classified as positive for respiratory, urinary and uncertainty, but negative for all other categories. Hence the classification tasks results in a multi-label classification. We have clarified this in the text, and avoiding using class (using category instead throughout).

9. L. 157 – reference to Appendix S7 – Labelling coverage – needs clarification – what is the line that goes beyond the 4000 set size – if only 4000 were labeled – is it prediction? Doesn't look like it. And what is ‘sample’ in the figure legend. Please, revise to make it clear.

We have updated the plot and legend, to show that this is a hypothetical labelling coverage of the dataset. If we label the most commonly occurring 4000 unique text strings (samples), we would achieve a 84% coverage. If we were to label 10 000, we would have a coverage of 90%

10. Also, it would be good to have more examples of various text-strings as a supplemental file.

We added a table showing the most common ten indication text strings to the appendix, alongside the existing table of examples of less common indications.

11. Figure 1 needs better Legend with exact description of what authors mean by ‘global’ and ‘distinct’; also, panels A and B would look better if they had categories in same order – easier to compare side-by-side.

We added an explanation to the legend explaining global and distinct (i.e., scaled according the prevalence of each string in the whole dataset and based just unique text strings).

12. Figure 2 and Methods/Results – ‘incomplete’ – example what is meant by it?

We added a description for incomplete data to the legend. It refers to text strings extracted from antibiotic prescriptions which can't be parsed/have no meaning. Examples would be blank strings, "NA", only numbers, or just punctuation excluding the question mark.

13. When comparing BERT model's performance to other models, authors use word 'substantially better' (L. 231) – is there a way to measure whether this performance is different significantly?

We have now added the overall accuracy scores for both models, to show that the "substantially better" improvement refers to the increase in accuracy from 0.14 -> 0.94.

14. AL. 232 – some comment or reference regarding the statement that regex 'might be chosen by many health institutions at present' would be nice.

We quantified the statement by explaining, that regex "might have been previously chosen by many health institutions" due to the ease of implementation.

Reviewer #3 (Remarks to the Author):

The paper focuses on automated classification and structuring of the information in clinical free text reports, with a focus on infection. From my understanding, the classification task focuses primarily on classifying individual infection indication terms that occur in infection reports. The study mainly focuses on comparing a set of machine learning models for this classification task.

As a reviewer, my expertise is mainly in machine learning, so it is difficult for me to say too much about the clinical novelty and relevancy of this work. From a machine learning perspective, there is no novelty in the methods used. Still, the comparison of classification methods/models for this task could be of some value to the health informatics community. Currently this paper lacks clarity in its explanations of how the data is prepared and how the classification task is defined and conducted. There are also some seemingly odd choices to be found regarding how the classification models are used, especially when it comes to the BERT model. Finally, although I might be mistaken and, the task seems to be relatively easy from a machine learning perspective. The variation to expect in the set of units to be classified (individual infection terms) is likely somewhat limited for many of the classes, this is also one reason why the performance scores are relatively high (close to 1 for several performance metrics).

We agree this a novel application of established machine learning methods on medical data, and therefore of most value to the health informatics community. Current literature suggests that the field is commonly using ICD-10 codes to determine infection sources for downstream analysis. In this study, we compare established ICD-10 codes to infection sources extracted from free-text and show that they offer more granularity – wider use of free-text using our approach could improve

medical research and service planning and delivery.

When I started to read the paper, I was quite sure you were focusing on several classification tasks, such as prescribed drugs, prescribing specialties, and infection indications/syndrome. For a time I was also certain that you were performing some kind of named entity recognition (NER, the task of identifying and labeling word sequences within sentences). However, when looking at the results, it seems that this is not the case. I suggest that you go through the paper and try to be more consistent in how you describe the task in the different sections.

In responding to all the reviewers, we have tried to improve the consistency of how the task is described, simplifying language where possible, and also providing additional examples, as described below. When have clarified the task we predict for in the abstract and start of the methods to try to avoid any confusion.

“... each antibiotic indication text string ...” – it is currently not fully clear to me what an antibiotic indication text string/term is. Please describe in detail how these are defined, identified and extracted for the manual labeling and later for the automated classification. Some of your examples seem to be “urinary/chest” and “sepsis ?source”.

We define antibiotic indication text string as the raw input for our model and classification task. It is the free-text that has been recorded alongside antibiotic prescriptions. We have now clarified that indication is reason the antibiotic is given early in the manuscript too.

Continuing the above – Is it so that each of these indicators are classified individually and out of context? This needs to be specified in more detail. I think a figure (occurring early in the main article) with a clear illustrative example based on a single (anonymized or made up) report would be very helpful for enable the reader to understand the classification task(s?), the units of classification, and how these are labeled.

These indications are indeed classified individually and separate from other entries in the EHR database. We have added a passage to the manuscript to explain this.

The task, as I understand it so far, is to classify the individual indication term(?). If my understanding is correct, I am tempted to believe that this set of unique terms is actually more or less finite (+ typos), at least for several of the classes, and within the same unit. This also explains why the scores are close to 1 for the different metrics for the best performing methods/models. It would be nice to see a discussion around the nature of the data and the task in the paper. I assume you are also interested in infection detection on the level of individual patients. Do you think the current performance scores looks the same on that level?

We agree, we have expanded our discussion, that there are a finite number of commonly used terms, which allows for such good performance. However, the use of LLMs and tokenisation still have the benefit of detecting subtle variation of

indications and permitting greater misspellings compared to a regex-based approach.

We also investigated the per-patient/per-sample accuracy which is reported as “overall accuracy” to reflect the accuracy of the entire per-sample/per-patient prediction across all eleven infection categories. We show that the finetuned Bio+Clinical BERT is able to classify 94% of all samples correctly – reflecting similar performance to the per class scores.

“An additional variable was used to document the presence of uncertainty expressed by the prescriber” – This requires more explanation. Are these somehow used in the classification task(s)? Consider also incorporating this in the requested example figure.

In addition to the infection sources (anatomical locations) we also added an additional category to reflect the uncertainty. It was added to dataset and training as an additional outcome in the multi-label classification task. We added the entry “Prescribing Uncertainty” to our glossary to explain the concept.

“Comparator Classification by ICD10 Codes” – It is not clear what these are used for. Are these used to generate labels for an additional classification task perhaps? Or maybe some additional reference data? If not, what are these used for? And, can you assume that these are available at classification time? Please clarify.

We are using ICD-10 codes here to benchmark the infection sources extracted from free-text, they are not used in our model. We added a clarifying passage to our manuscript and reordered the methods to make this clearer.

As the reviewer mentions, ICD-10 are usually not available at the time of the classification and therefore pose as a limitation compared to sources extracted from free-text for applications where data on the source of infection are used in real-time. However, this is less of a concern for retrospective analyses.

“(details below)” – Please replace these with more specific pointers.

We have updated the sections with more specific pointers to the referred paragraph.

“Few-Shot and finetuned LLM Classifier” – Please provide more details how generative LLMs are used to classify a report. Why not show how the output looks like in Appendix S3?

The output in Appendix S2 (previously S3) refers to the output of both the Zero-Shot (corrected from Few-shot) and finetuned LLM. We now also added the output JSON and changed the section title to reflect this.

Continuing the above - Did you perhaps encounter any problems of unexpected outputs from these? If so, how were these handled?

We experienced unexpected output as described by the reviewer at the beginning of our study. If the models were not forced to return a JSON formatted output, they potentially make up new categories. The prompt includes multiple instructions to only return the classes specified in the prompt. We added the rationale behind choosing our prompt to the description in the Appendix S2.

“BERT Classifier” – It is a bit unclear how the data is presented to the model and how you performed the classifications. Are you perhaps applying the classification layer to the [CLS] token? If so, does this mean you give the model one indication string at a time to classify? BERT excels at contextualized classifications; thus, this sounds like an unused potential. Why not give the full report to the model and perform multi-label classification? You may or may not want to first remove any ‘non-indication terms’. If you need to classify the individual tokens, one option would be to classify each token in a NER-CRF based manner with BERT (e.g., see TFAutoModelForTokenClassification in the Huggingface library).

As we have clarified above (and in the manuscript too), that the model only takes a small portion of text, with no context. Therefore, the context described in this comment, does not apply as originally intended.

Table 1 – I suggest that you report both Micro and Macro scores for the various metrics. This will provide a better insight into the classification performances (which may or may not be influenced by label imbalances).

We added a table with weighted, micro and macro average scores where applicable to our appendix, to allow for comparisons across different averaging methods.

30th October 2024

Dear Reviewers,

Re: COMMSMED-24-0650A

Thank you for your feedback and the opportunity to respond to the remaining reviewer suggestions, which we do below.

Best wishes,

Kevin Yuan and David Eyre on behalf of all authors

Reviewer #3 (Remarks to the Author):

Thank you for the replies to my comments and questions. Most of my questions and concerns have been addressed. It is now quite clear how the task is formulated and what the unit of classification is, and the experiments and evaluations look good. Here are my comments:

Could you clarify whether or not you performed some text preprocessing of the indication texts? You mention at least some filtering being done the caption of Figure 2. E.g., did you normalize the text in some way, remove some special characters, and/or did you perhaps apply some (sentence) splitting to potentially split the original indication text (from the free-text box) into multiple indication text units to be classified? If any such preprocessing was used, I think it would be good to clarify this in the manuscript.

We converted the indications to lower case to reduce the number of variants the same string as there often was inconstant use of casing. We did not apply any further pre-processing steps such as splitting the original indication text into multiple units, as the tokenisers for XGBoost and BERT took care of that step.

We added a subsection to the appendix to explain the pre-processing and filtering steps we undertook to create the dataset.

In Table S8.2 - You probably meant "Common" instead of "Uncommon".

We thank the reviewer for catching the typo; the column header has been updated to "Common Indication."

I am somewhat wondering if the term/phrase "feature extractors" ("feature extraction") is really representative for the task and experiments that you report here. I guess it is ok, but you could give it some thought.

We have consider the feedback, and feel that it is a reasonable description of the task that can be understood by those with domain knowledge relevant to language processing and also a more general audience.

Finally, as mentioned in the previous review round, I think that the comparison of these methods at this classification task is valuable and interesting.

Thank you.